# Transcriptome Expression Profiling Reveals the Molecular Response to Salt Stress in *Gossypium anomalum* Seedlings

**DOI:** 10.3390/plants13020312

**Published:** 2024-01-20

**Authors:** Huan Yu, Qi Guo, Wei Ji, Heyang Wang, Jingqi Tao, Peng Xu, Xianglong Chen, Wuzhimu Ali, Xuan Wu, Xinlian Shen, Yinfeng Xie, Zhenzhen Xu

**Affiliations:** 1Co-Innovation Centre for Sustainable Forestry in Southern China, College of Life Sciences, Nanjing Forestry University, Nanjing 210037, China; yuhuan07112021@163.com; 2Key Laboratory of Cotton and Rapeseed (Nanjing), Ministry of Agriculture and Rural Affairs, The Institute of Industrial Crops, Jiangsu Academy of Agricultural Sciences, Nanjing 210014, China; sevenguo86@163.com (Q.G.); skye0622@163.com (W.J.); whywhy0325@163.com (H.W.); tjq991116@163.com (J.T.); semon528@hotmail.com (P.X.); jshzlcxl@jiuheseed.com (X.C.); 15729797074@163.com (W.A.); amazingjackey@163.com (X.W.); xlshen68@126.com (X.S.)

**Keywords:** *Gossypium anomalum*, salt stress, physiological and biochemical index, RNA-Seq, molecular response

## Abstract

Some wild cotton species are remarkably tolerant to salt stress, and hence represent valuable resources for improving salt tolerance of the domesticated allotetraploid species *Gossypium hirsutum* L. Here, we first detected salt-induced stress changes in physiological and biochemical indexes of *G. anomalum*, a wild African diploid cotton species. Under 350 mmol/L NaCl treatment, the photosynthetic parameters declined significantly, whereas hydrogen peroxide (H_2_O_2_) and malondialdehyde (MDA) contents increased. Catalase (CAT), superoxide dismutase (SOD), and peroxidase (POD) activity and proline (PRO) content also significantly increased, reaching peak values at different stages of salt stress. We used RNA-Seq to characterize 15,476 differentially expressed genes in *G. anomalum* roots after 6, 12, 24, 72, and 144 h of salt stress. Gene Ontology enrichment analysis revealed these genes to be related to sequence-specific DNA and iron ion binding and oxidoreductase, peroxidase, antioxidant, and transferase activity; meanwhile, the top enriched pathways from the Kyoto Encyclopedia of Genes and Genomes database were plant hormone signal transduction, phenylpropanoid biosynthesis, fatty acid degradation, carotenoid biosynthesis, zeatin biosynthesis, starch and sucrose metabolism, and MAPK signaling. A total of 1231 transcription factors were found to be expressed in response to salt stress, representing ERF, MYB, WRKY, NAC, C_2_H_2_, bZIP, and HD-ZIP families. Nine candidate genes were validated by quantitative real-time PCR and their expression patterns were found to be consistent with the RNA-Seq data. These data promise to significantly advance our understanding of the molecular response to salt stress in *Gossypium* spp., with potential value for breeding applications.

## 1. Introduction

Soil salinization is one of the most pressing ecological and environmental challenges faced by the world today, with increasing soil salinity having caused a significant reduction in arable land worldwide. The Food and Agriculture Organization of the United Nations (FAO) reports that there are more than 833 million hectares of salt-affected soil around the globe (8.7% of the planet) [1]. Both soil salinity and scarcity of freshwater supplies have severely hindered global agricultural growth, highlighting the cash crop–food crop dilemma. Cotton (*Gossypium* spp.), a significant source of natural fiber for the textile industry [2], can grow in moderate salinity areas. After 5–10 years, the improved land can then be used for growing wheat, corn, and other crops [3,4]. Thus, to ensure safe and stable food production, cotton production areas are continually moving to saline land that is unsuitable for some food crops [5]. This process is essential for expanding the amount of arable land for food production and addressing the conflict between land use for food and cotton, thus ensuring food security.

With a relatively high yield potential and medium fiber quality, *Gossypium hirsutum* L. (known as upland cotton), a domesticated allotetraploid species (2n = 4x = 52, (AD)_1_), accounts for almost 90% of the world’s cotton production annually [6]. Although it is fairly tolerant to salt stress, upland cotton is still subject to salt-induced impairment of growth, productivity, and fiber quality, especially during the germination, emergence, and seedling stages [7]. Artificial selection has unintentionally narrowed the genetic variability of modern upland cotton cultivars, and makes it challenging for scientists to further improve their salt tolerance. Wild cotton species mostly grow in semi-arid and subtropical coastal islands, and hence not only have a strong adaptability to drought, but also can tolerate the invasion of sea tides. During long-term adaptive evolution, some wild cotton species may have come to possess morphological, physiological, and molecular mechanisms for salt tolerance, as well as valuable alleles that can be used to improve the salt tolerance of modern cultivated upland cotton [8,9]. 

*G. anomalum*, a wild African diploid cotton species belonging to the B_1_ genome group, thrives in the southwest of Africa and along the fringes of the Sahara Desert [10]. Due to its adaptation to harsh environments, plants of this species exhibit morphological phenotypes that increase tolerance to abiotic stress, such as small and deeply parted palmate leaves, thick and hard seed coats, and dense hairs. In addition, improvement of a plant’s resistance to stress is closely tied to the effectiveness of its antioxidant enzyme system [11,12]. Therefore, *G. anomalum* is a valuable wild species for increasing our understanding of the mechanism and biological functions of genes related to abiotic stress in *Gossypium* spp. In general, when plants experience salt stress during their growth, they produce a significant amount of reactive oxygen species (ROS), which can negatively impact cellular function by damaging nucleic acids and oxidizing proteins. Furthermore, peroxidation of membrane lipids, particularly in the plasma membrane, damages the structure of the cell membrane and alters its permeability. As a byproduct of this process, malondialdehyde (MDA) is produced. To counteract the stress-induced accumulation of ROS, plants employ signal transduction and an antioxidant enzyme system that includes enzymes such as superoxide dismutase (SOD), catalase (CAT), and peroxidase (POD). To date, there is no transcriptome available for salt-stressed *G. anomalum* in the NCBI database (https://www.ncbi.nlm.nih.gov/, accessed on 21 December 2023), and hence no data on its expression of antioxidant system genes in relation to salt exposure. Here, we first detected the impact of salt stress on physiological and biochemical indexes in *G. anomalum* seedlings. Then, we used RNA-Seq to characterize the main regulators and pathways and molecular mechanism of salt tolerance in *G. anomalum* roots after 6, 12, 24, 72, and 144 h of 350 mmol/L NaCl treatment. These data will promise to significantly advance our understanding of the molecular response to salt stress in *Gossypium* spp. and may be valuable for breeding application.

## 2. Results

### 2.1. Physiological and Biochemical Evaluation of G. anomalum Seedlings Response to Salt Stress

Under 350 mmol/L NaCl treatment, *G. anomalum* seedlings continued to grow, and leaf color and performance after 6 h, 12 h, 24 h, and 72 h of salt stress were not significantly different from the 0 h time point (Figure 1A). Only after 144 h did cotyledons and true leaves show slight curling and dehydration (Figure 1A).

The net photosynthetic rate (Pn, μmol m^−2^ s^−1^), stomatal conductance (Gs, μmol m^−2^ s^−1^), transpiration rate (Tr, mol m^−2^ s^−1^), and intercellular CO_2_ concentration (Ci, µmol mol^−1^) were significantly decreased (*p* < 0.05, *p* < 0.01) in salt-treated *G. anomalum* plants compared to the 0 h control (Figure 1B–E). Notably, Gs, Tr, and Ci all showed slight increases at 72 and 144 h compared to 24 h, while Pn exhibited a decreasing trend throughout the whole salt stress period (Appendix A).

To detect the degree of oxidative and membrane damage caused by salt stress, H_2_O_2_ and MDA contents were quantified in salt-treated *G. anomalum* plants. In the leaves, the relative MDA content showed a significant increasing trend (*p* < 0.05, *p* < 0.01), reaching a peak at 72 h (Figure 2A). Meanwhile, the H_2_O_2_ content did not change after 24 h of salt treatment, but significantly increased at 72 h and 144 h (*p* < 0.05, *p* < 0.01) (Figure 2B). Most notably, both MDA and H_2_O_2_ contents showed a slight decline at 144 h (Figure 2A,B). This corresponded with significant salt-stress-related increases (*p* < 0.05, *p* < 0.01) in antioxidant enzyme activities, including CAT, SOD, and POD (Figure 2C–E). POD and SOD activity reaching their highest values at 72 h of salt stress, whereas CAT peaked at 24 h (Figure 2C–E). Finally, as the osmoregulatory substance (proline (PRO) content) showed a significant increase at 24 h of salt stress, and a continued increasing trend with further treatment, reaching its highest value at 144 h (Figure 2F).

### 2.2. Analysis of RNA-Seq Data

To identify genes associated with salt tolerance, RNA-Seq was conducted on *G. anomalum* roots after 0, 6, 12, 24, 72, and 144 h of 350 mmol/L NaCl treatment. Three replicates were performed for each treatment. Each sample generated at least 39,807,348 raw reads (Appendix A). These raw reads were subjected to quality inspection and adapter removal, resulting in 39,762,028 to 49,053,344 high-quality clean reads (Appendix A). Read quality was assessed by calculating the Q20 and Q30 values, which were ≥97.29% and 92.41%, respectively (Appendix A). Of the total clean reads, approximately 92.79% to 96.78% were mapped to the reference sequence of *G. anomalum* [13], and 90.17% to 94.44% of those mapped reads were uniquely aligned (Appendix A). Additional de novo assembly of all valid clean reads resulted in 3552 novel contigs.

Differentially expressed genes (DEGs) in *G. anomalum* roots relative to the 0 h control were identified at 6, 12, 24, 72, and 144 h of treatment with 350 mmol/L NaCl. The screening criteria were based on transcript expression abundance (|log2FoldChange| ≥ 1 and *P*adj ≤ 0.05), and yielded a total of 15,476 DEG (Appendix A). Concerning each timepoint, 5699 (3688 up-regulated/1981 down-regulated), 9025 (5262 up-regulated/3763 down-regulated), 4981 (2702 up-regulated/2279 down-regulated), 5166 (2478 up-regulated/2688 down-regulated), and 8249 (3670 up-regulated/4579 down-regulated) DEGs were detected at 6, 12, 24, 72, and 144 h of salt treatment, respectively (Figure 3A and Appendix A). The greatest number of up-regulated genes was observed with 6 h of salt stress, while the greatest number of down-regulated genes was seen at 144 h (Figure 3A). Clustering of the expression patterns of these 15,476 DEGs revealed the 6, 12, 24, and 72 h samples to be closely grouped, representing the early stage of salt stress, while the 144 h sample was separately grouped, representing the late stage (Figure 3B)

### 2.3. Gene Ontology (GO) Annotation and Enrichment Analysis

To elucidate the biological processes influenced by salt stress in *G. anomalum*, Gene Ontology (GO) annotation was conducted for the 15,476 DEGs, including terms from the biological process (BP), cellular component (CC), and molecular function (MF) ontologies. A total of 7656 DEGs were annotated to 42 functional terms, including 16 BP, 11 CC, and 15 terms (Figure 4 and Appendix A); a considerable proportion of DEGs (7820, 50.52%) did not map to any term. In the BP category, the most frequently annotated terms were metabolic process (GO:0008152), cellular process (GO:0009987), single-organism process (GO:0044699), biological regulation (GO:0065007), localization (GO:0051179), response to stimulus (GO:0050896), and cellular component organization or biogenesis (GO:0071840) (Figure 4 and Appendix A). Among the CC terms, the most highly represented were cell part (GO:0044464), cell (GO:0005623), membrane (GO:0016020), and membrane part (GO:0044425) (Figure 4 and Appendix A). Finally, within the MF category, the most prevalent terms were binding (GO:0005488), catalytic activity (GO:0003824), nucleic acid binding transcription factor activity (GO:0001071), transporter activity (GO:0005215), and structural molecule activity (GO:0005198) (Figure 4 and Appendix A).

Subsequent enrichment analysis identified 55 enriched GO terms overall (*P*adj ≤ 0.05), including 40 MF, 11 BP, and 4 CC terms (Figure 5A–C and Appendix A). Among the MF category, the enriched terms included sequence-specific DNA binding (GO:0043565), peroxidase activity (GO:0004601), oxidoreductase activity (GO:0016684), antioxidant activity (GO:0016209), microtubule motor activity (GO:0003777), iron ion binding (GO:0005506), acyltransferase activity (GO:0016747), microtubule binding and motor activity (GO:0008017), hydrolase activity, hydrolyzing O-glycosyl compounds (GO:0016798), and acyltransferase activity (GO:0016746) (Figure 5A and Appendix A). Of the BP terms, the greatest enrichments were seen for polysaccharide metabolism (GO:0005976), response to oxidative stress (GO:0006979), cellular or subcellular component movement (GO:0006928), microtubule-based movement (GO:0007018), cellular carbohydrate metabolic process (GO:0044262), cellular glucan metabolic process (GO:0006073), and glucan metabolic process (GO:0044042) (Figure 5B and Appendix A). Finally, there were only four enriched CC terms: cell wall (GO:0005618), extracellular organelle (GO:0030312), cytoplasm (GO:0048046), and extracellular region (GO:0005576) (Figure 5C and Appendix A).

### 2.4. Kyoto Encyclopedia of Genes and Genomes (KEGG) Annotation and Enrichment Analysis

Of the 15,476 DEGs, 2698 were annotated to 125 metabolic pathways in the Kyoto Encyclopedia of Genes and Genomes (KEGG) database, including 3193 DEGs in metabolism, 458 in genetic information processing, 465 in environmental information processing, 153 in cellular processes, and 179 in organismal systems pathways. Because a gene may be involved in multiple pathways, the sum of DEG counts over all the pathways is greater than the number of genes annotated (Figure 6 and Appendix A). Because a gene may be involved in multiple pathways, the sum of DEG counts over all the pathways is greater than the number of genes annotated (Figure 6 and Appendix A). Of the DEGs implicated in metabolism, most had functions relating to carbon metabolism (ko01200), phenylpropanoid biosynthesis (ko00940), biosynthesis of amino acids (ko01230), biosynthesis of cofactors (ko01240), starch and sucrose metabolism (ko00500), glycolysis/gluconeogenesis (ko00010), cysteine and methionine metabolism (ko00270), amino sugar and nucleotide sugar metabolism (ko00520), and pyruvate metabolism (ko00620) (Figure 6 and Appendix A). For those involved in genetic information processing, a large portion were annotated as contributors to protein processing in the endoplasmic reticulum (ko04141), ubiquitin mediated proteolysis (ko04120), ribosome biogenesis in eukaryotes (ko03008), RNA degradation (ko03018), spliceosome (ko03040), DNA replication (ko03030), mRNA surveillance pathway (ko03015), and nucleocytoplasmic transport (ko03013) (Figure 6 and Appendix A). There were only three impacted pathways in environmental information processing, including plant hormone signal transduction (ko04075), MAPK signaling pathway–plant (ko04016), and ABC transporters (ko02010) (Figure 6 and Appendix A). Among the cellular processes, the four impacted pathways were endocytosis (ko04144), peroxisome (ko04146), phagosome (ko04145), and autophagy–other (ko04136) (Figure 6 and Appendix A). Finally, only two organismal systems pathways harbored DEGs: plant–pathogen interaction (ko04626) and circadian rhythm (ko04712) (Figure 6 and Appendix A).

Enrichment analysis of the DEGs identified 24 important KEGG pathways. We constructed scatter plots to visualize these enrichments (Figure 7). The top enriched pathways were plant hormone signal transduction (ko04075), phenylpropanoid biosynthesis (ko00940), fatty acid degradation (ko00071), carotenoid biosynthesis (ko00906), zeatin biosynthesis (ko00908), starch and sucrose metabolism (ko00500), and MAPK signaling pathway (ko04016) (Figure 7 and Appendix A). 

### 2.5. Transcriptional Factors (TFs) with Altered Expression Response to Salt Stress

Across all the treatment timepoints (6, 12, 24, 72, and 144 h), a total of 1231 transcription factors (TFs) were differentially expressed when compared with the control (0 h). These TFs belonged to 50 different families, of which the ERF family showed the most significant alterations under salt stress, followed by the MYB, WRKY, bHLH, NAC, C_2_H_2_, bZIP, and HD-ZIP families (Figure 8 and Appendix A).

### 2.6. Validation of RNA-Seq Data by qRT-PCR

Based on the expression level in the transcriptome data and functional annotations from their orthologues in *Arabidopsis*, nine candidate genes potentially implicated in salt tolerance were selected to be used in quantitative real-time PCR (qRT-PCR). Relative transcript levels of these genes were computed using the 2^−∆Ct^ method, where ∆Ct is the difference of the Ct values between the target gene products and an internal standard gene product. These nine selected genes comprised cytochrome P450 (*Goano03G0241* and *Goano12G2738*), a WRKY transcription factor (*Goano03G1831*), serine acetyltransferase (*Goano01G2022*), a transferase family protein (*Goano05G2006*), a protein with a peroxidase structure domain (*Goano08G1049*), cellulose synthase (*Goano07G2681*), a 1,3-beta-glucan synthase component (*Goano05G1803*), and a homeobox-leucine zipper protein (*Goano02G0334*). The expression patterns of these nine genes as determined by qRT-PCR were consistent with the RNA-Seq data (Figure 9 and Appendix A).

## 3. Discussions

The changing climate is intensifying salt stress globally. Exposure of plants to salt stress results in many effects, including osmotic and toxic effects as well as oxidative stress and inhibition of photosynthesis [14,15,16]. In this study, photosynthetic parameters, Pn, Gs, Tr, and Ci all demonstrated a significant decline in salt-stressed *G. anomalum* seedlings. These results indicate that salt stress significantly inhibits the photosynthetic function of *G. anomalum* leaves. Broadly, salt stress is known to result in the disruption of photosynthetic apparatus and activities, with consequently reduced photosynthetic parameters [17,18]. At the late stress stage (72 h and 144 h), Gs, Tr, and Ci were slightly increased, reflecting a slight recovery of photosynthetic activity. This was in agreement with the appearance of visual symptoms of salt stress in that *G. anomalum* seedlings grew gradually and exhibited good performance after 6, 12, 24, and 72 h of salt stress, only showing slight curling and dehydration at 144 h. Thus, like other wild *Gossypium* species, such as *G. aridum* [19], *G. klotzschianum* [20], and *G. davidsonii* [21], *G. anomalum* showed tolerance to salt stress, even at the high NaCl concentration applied here (350 mmol/L). Identifying key genes involved in *G. anomalum* salt response will lay a solid foundation for further understanding the molecular mechanisms for salt tolerance in *Gossypium* spp. generally.

Salt stress can induce ROS, which leads to secondary oxidation stress, disturbs cellular redox homeostasis, and damages cell component and structure. H_2_O_2_, as one of the most abundant ROS in cells, is a key signaling molecule for plant growth, development, and resistance to stress [22]. In this study, we observed that the H_2_O_2_ content was significantly elevated under salt treatment, but slightly decreased at 144 h. Notably, MDA content, another stress-related physiological parameter, also demonstrated a significant increasing trend under stress with a slight decline at 144 h. These results suggested that early stages of salt stress in *G. anomalum* featured severe oxidative and membrane damage, but alleviation of oxidative damage occurs in the late stress stage. Plants possess complex antioxidative defense systems by which they maintain reactive ROS scavenging ability and control intracellular homeostasis [23,24,25], and alleviation of oxidative ROS damage and improvement of salt resistance is often correlated with the balance between ROS producing and scavenging [26]. In this study, the CAT, SOD, and POD activity were all significantly increased with salt stress exposure, achieving peak values at 24 h, 72 h, and 72 h, respectively, suggesting that these enzymes are called upon in their respective roles during different stages of salt stress. Also, we found salt-stress-induced DEGs to be significantly enriched for GO terms related to oxidoreductase activity (GO:0016684 and GO:0016705), peroxidase activity (GO:0004601), and antioxidant activity (GO:0016209).

Numerous studies have also suggested that mitogen-activated protein kinases (MAPKs) are involved in ROS homeostasis [27,28]. Several genes encoding MAPKs, such as *GhMPK2*, *GhMPK5*, and *GhMPK17,* were cloned in *G. hirsutum* L. and found to positively regulate stress tolerance [29,30,31]. In this study, the MAPK signaling pathway-plant (ko04016) was one of the top enriched pathways among the 15,476 DEGs. Moreover, MAPK-related genes, such as *MKK2* (*Goano07G0148*), *MPK4* (*Goano08G1893*), *MPK6* (*Goano01G1166*), and *MPKKK17_18* (*Goano10G0114*, *Goano00G0189*, *Goano01G2100*, and *Goano05G1918*), were up-regulated throughout almost the entire salt stress stage treatment. In addition, the DEGs exhibited significant enrichment for the GO terms of sequence-specific DNA binding (GO:0043565), iron ion binding (GO:0005506), and transferase activity (GO:0016747), findings that are in agreement with previous studies [32,33]. To date, no gene related to salt tolerance has been cloned in *G. anomalum*. Cloning these potential elite salt tolerance genes and overexpressing them in *G. hirsutum* will assist in the development of new upland cotton cultivars with improved salt tolerance.

Plant hormones, especially abscisic acid (ABA), play crucial roles in the response to salt stress by regulating plant growth and development adaptation [34]. In the absence of ABA, the kinase activity of SnRK2 is inhibited by protein phosphatase 2Cs (PP2Cs) [35]. In response to salinity and osmotic stress, plants rapidly increase endogenous ABA levels, with consequent enhancement of ABA signaling. ABA receptor proteins (pyrabactin resistance, PYR) and the regulatory component of ABA receptors (RCAR) perceive and bind to the accumulated ABA, an interaction that subsequently inhibits the phosphatase activity of PP2Cs [35,36,37], followed by the rapid activation of SnRK2. In the present study, the most enriched pathway among the DEGs was plant hormone signal transduction (ko04075). Notably, seven genes related to PYR/PYL (*Goano08G3377*, *Goano06G2297*, *Goano05G0781*, *Goano05G2277*, *Goano04G2489*, and *Goano09G1265*) and SnRK2 (*Goano04G2494*, *Goano00G0189*, and *Goano05G1918*) were up-regulated in salt stress *G. anomalum*, which heavily implicates ABA in the response to salt stress. 

Transcription factors are the main master regulators that control gene clusters involved in ABA-dependent and ABA-independent pathways [38]. Under salt treatment, the increased level of ABA promotes the activity of downstream TFs that modulate the expression of ABA-responsive genes [39,40]. In our data, 1231 TFs were differentially expressed under salt stress; among them, the most highly represented families were ERF, MYB, WRKY, and bHLH, which could regulate salt tolerance through ABA signaling and modulation of ROS production [41,42]. Previous studies have illustrated that some TFs are essential in the salt response mechanism of cotton. For instance, the silencing of two ERF members, *GhERF4L* and *GhERF54L*, significantly reduced the salt tolerance of cotton seedlings [43], while the expression of *GhWRKY3* can be induced in *G. hirsutum* L. by NaCl treatment [44]. Overexpression of *GarWRKY5* in *Arabidopsis* was shown to positively regulate salt tolerance during seed germination and vegetative growth [45]. Likewise, the cotton MYB gene, *GhMYB73,* was found to be induced by NaCl and ABA, and its overexpression significantly increased tolerance to salt and ABA stress [46]. Further analysis of the salt-responsive TFs identified in this study will be very useful as a direction for the genetic improvement of salt resistance in cultivated upland cotton.

## 4. Materials and Methods

### 4.1. Preparation of Plant Material

Seeds of G. *anomalum* were soaked in distilled water for 24 h, and then were pre-germinated with distilled water in a plant incubator at 60% humidity and a 16 h day (8 °C)/8 h night (23 °C) cycle. Afterwards, the germinated seeds were planted in soil and cultured in the same conditions. After 15 days, the seedlings possessed two leaves and one heart. Uniformly developed seedlings were selected for salt treatment. Every plant was placed in a paper cup (diameter 7 cm and height 8 cm) and irrigated with 75 mL of 350 mmol/L NaCl [47]. Roots were collected at 0, 6, 12, 24, 72, and 144 h after salt treatment. These samples were immediately frozen in liquid nitrogen and stored at −80 °C for RNA extraction.

### 4.2. Evaluation of Physiological and Biochemical Indexes

Photosynthetic parameters (Tr, Pn, Ci, and Gs) of the test plant materials were measured with the Li-6800 photosynthetic analyzer (Li-Cor Inc., Lincoln, NE, USA). The air relative humidity and photosynthetically active radiation were set to 50% and 1500 µmol m^−2^ s^−1^, respectively, and the CO_2_ concentration in the reference leaf chamber was controlled at a value of 400 µmol mol^−1^. In each biological replicate, the second leaf from the top was observed for five individual seedlings.

Membrane lipid peroxide content (MDA and H_2_O_2_), PRO, and oxidoreductase activity (CAT, POD, and SOD) were likewise measured in the second leaf from the top of five individual seedlings at 0, 6, 12, 24, 72, and 144 h after 350 mmol/L NaCl treatment; all were determined using appropriate assay kits (Jiancheng, Nanjing, China).

### 4.3. Library Preparation for RNA-Seq

Total RNA for each sample was extracted using a plant RNA purification kit (Omega, Beijing, China). RNA integrity and quantity were assessed using the RNA Nano 6000 detection kit of the Bioanalyzer 2100 system (Agilent Technologies, Santa Clara, CA, USA). Libraries were constructed using the Illumina TruSeq Stranded RNA Library Preparation Kit (Illumina, San Diego, CA, USA), and quantified using the Qubit2.0 fluorometer (Invitrogen, Carlsbad, CA, USA). All libraries were pooled according to the effective concentration and target amount of data, and were sequenced on the Illumina NovaSeq 6000 platform (150 bp paired-end).

### 4.4. Differential Expression Analysis

Reads containing adapters and low-quality raw data were detected and removed using the FastQC v0.11.9 software. The resulting high-quality clean reads were then aligned to the reference genome of G. *anomalum* [13] with the Hisat2 (v2.0.5) software [48]. In order to supplement the unannotated genes in the reference genome, valid clean reads from all samples were also assembled using the StringTie (v1.3.3b) software [49]. The expression of each gene was calculated using the featureCounts v1.5.0-p3 software. Differential expression analysis comparing the 6, 12, 24, 72, and 144 h libraries with the control (0 d) was performed using the *DESeq2* R package (1.20.0) [50,51]. After controlling the false discovery rate using Benjamini and Hochberg’s method, genes with |log2FoldChange| ≥ 1 and *P*adj ≤ 0.05 were identified as differentially expressed genes (DEGs).

### 4.5. GO and KEGG Annotation and Enrichment Analysis

For functional and pathway annotation, all DEGs were mapped to Gene Ontology (GO) and the Kyoto Encyclopedia of Genes and Genomes (KEGG) database, and the number of DEGs in each term and pathway was calculated. To identify significantly enriched terms and pathways, the DEGs were considered as the foreground, while the 42,752 genes in the *G. anomalum* genome [13] were used as the reference background. The *p*-value of the foreground DEGs in a specific term/pathway was calculated using the hypergeometric distribution algorithm (Phyper) and then corrected for the false discovery rate (FDR) [52].

### 4.6. Transcription Factor (TF) Annotation

Transcription factors among the DEGs were identified by BLASTx (version 5.0) comparison against the plant transcription factor database (PlnTFDB) [53].

### 4.7. Validation of DEGs by qRT-PCR

For RNA-Seq data validation, nine candidate genes that are potentially responsible for salt tolerance were chosen and their expression levels confirmed by quantitative real-time PCR (qRT-PCR). Endogenous cotton *histone-3* (*AF024716*) was used as an internal standard to determine the total amount of cDNA template. Gene-specific primers for the candidate genes were designed using the Snap Gene software (version 6.0.2). The TB Green^®^ Premix Ex TaqTM (Tli RNaseH Plus) kit (RR820B) from TaKaRa was utilized for nucleic acid synthesis. All reactions were run on the QuantStudio 5 real-time PCR system. The relative transcription level was calculated using the 2^−ΔCt^ method, in which ΔCt represents the difference in Ct values between the target gene products and the internal standard product (*histone-3*) [54].

## 5. Conclusions

Photosynthetic parameters in *G. anomalum* seedlings, including Pn, Gs, Tr, and Ci, declined significantly under salt stress, while H_2_O_2_ and MDA contents were increased. CAT, SOD, and POD activity and PRO content were all significantly increased after salt stress, reaching their highest values during different stress stages. RNA-Seq analysis of salt-treated *G. anomalum* plants yielded 15,476 DEGs. According to GO enrichment analysis, these DEGs were related to sequence-specific DNA binding, oxidoreductase activity, peroxidase activity, antioxidant activity, iron ion binding, and transferase activity; meanwhile, KEGG enrichment analysis highlighted plant hormone signal transduction, phenylpropanoid biosynthesis, fatty acid degradation, carotenoid biosynthesis, zeatin biosynthesis, starch and sucrose metabolism, and MAPK signaling pathway-plant. Finally, a number of TFs, including ERF, MYB, WRKY, and bHLH family members, were found to be induced in response to salt stress. These data will provide deeper insights into the molecular mechanism of salt stress adaptation in *Gossypium* spp. and might have valuable implications for breeding applications. 

## Figures and Tables

**Figure 1 plants-13-00312-f001:**
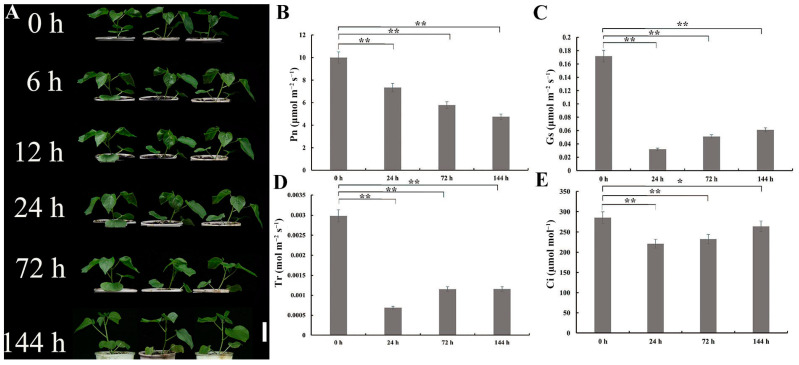
Phenotype and photosynthetic parameters of *G. anomalum* seedlings under salt treatment. (**A**) Phenotype of *G.anomalum* seedings under salt treatment. Scale bar = 10 cm. (**B**) Pn: net photosynthetic rate. (**C**) Gs: stomatal conductance. (**D**) Tr: transpiration rate. (**E**) Ci: intercellular CO_2_ concentration. * *p* < 0.05, ** *p* < 0.01, Student’s *t* test.

**Figure 2 plants-13-00312-f002:**
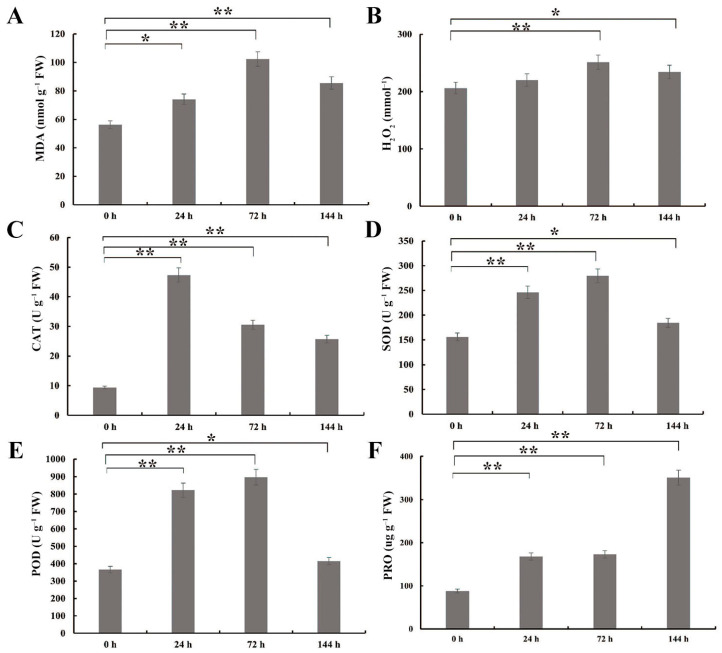
Physiological and biochemical indexes of *G. anomalum* seedlings under salt stress. (**A**) MDA: malondialdehyde. (**B**) H_2_O_2_: hydrogen peroxide. (**C**) CAT: catalase. (**D**) SOD: superoxide dismutase. (**E**) POD: peroxidase. (**F**) PRO: proline. * *p* < 0.05, ** *p* < 0.01, Student’s *t* test.

**Figure 3 plants-13-00312-f003:**
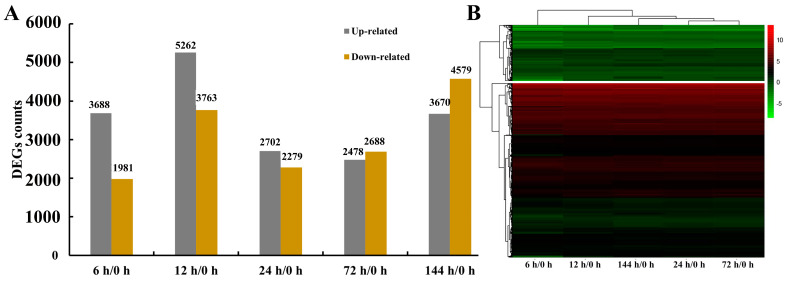
Counts of genes in *G. anomalum* under salt stress (**A**). Time points refer to the number of hours exposed to salt treatment. Hierarchical clustering of genes in *G. anomalum* under salt stress (**B**). Red indicates up-regulated genes and green represents down-regulated genes.

**Figure 4 plants-13-00312-f004:**
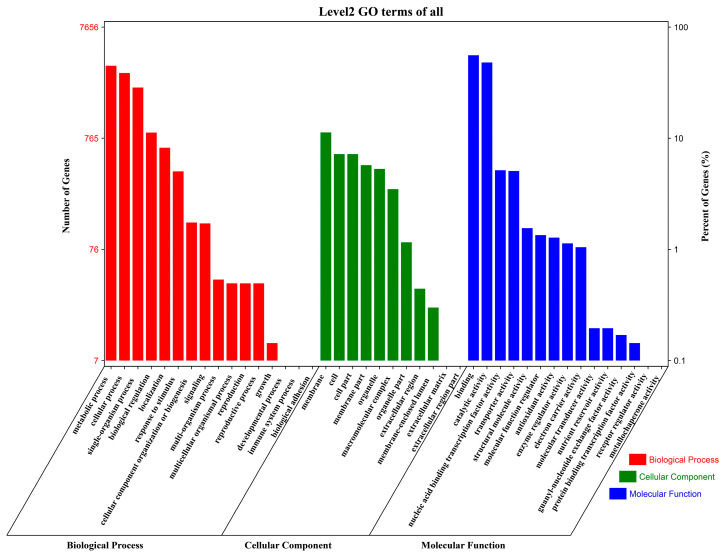
Gene Ontology (GO) annotation distribution of genes differentially expressed in *G. anomalum* under salt stress. Histograms represent the functional distribution; the left y-axis shows the number of annotated genes, and the right y-axis represents the percentage of total genes.

**Figure 5 plants-13-00312-f005:**
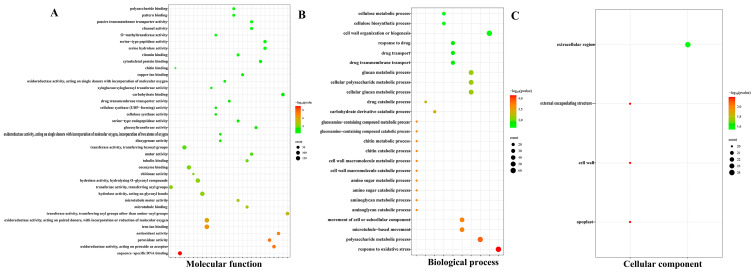
Gene Ontology (GO) enrichment analysis of genes differentially expressed in *G. anomalum* under salt stress. (**A**) Molecular function. (**B**) Biological process. (**C**) Cellular component. Color intensity indicates relative degree of significance; dot size indicates the number of DEGs annotated with the term. Only the top 30 significantly enriched terms are displayed for each category.

**Figure 6 plants-13-00312-f006:**
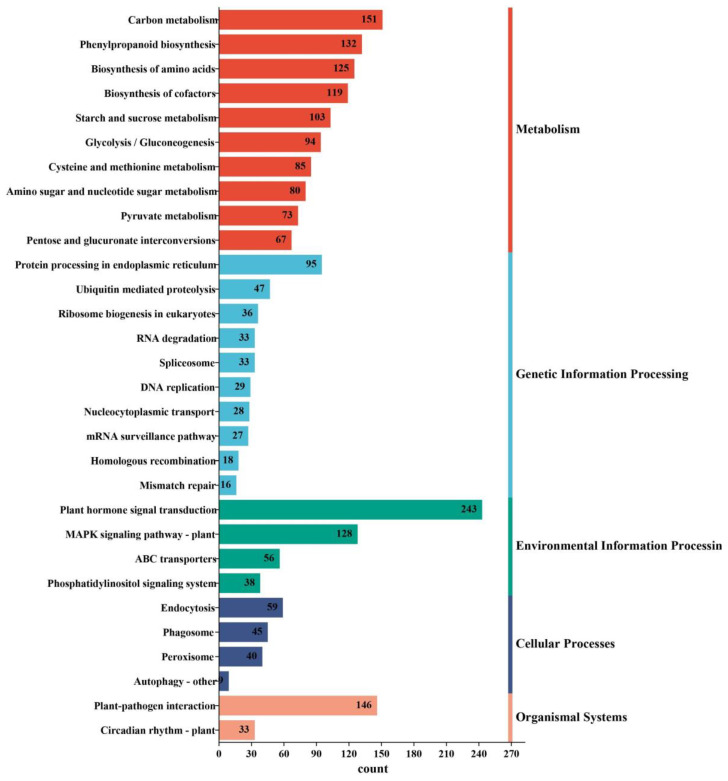
Kyoto Encyclopedia of Genes and Genomes (KEGG) annotation distribution of genes differentially expressed in *G. anomalum* under salt stress. Only the top ten pathway terms are displayed for each group.

**Figure 7 plants-13-00312-f007:**
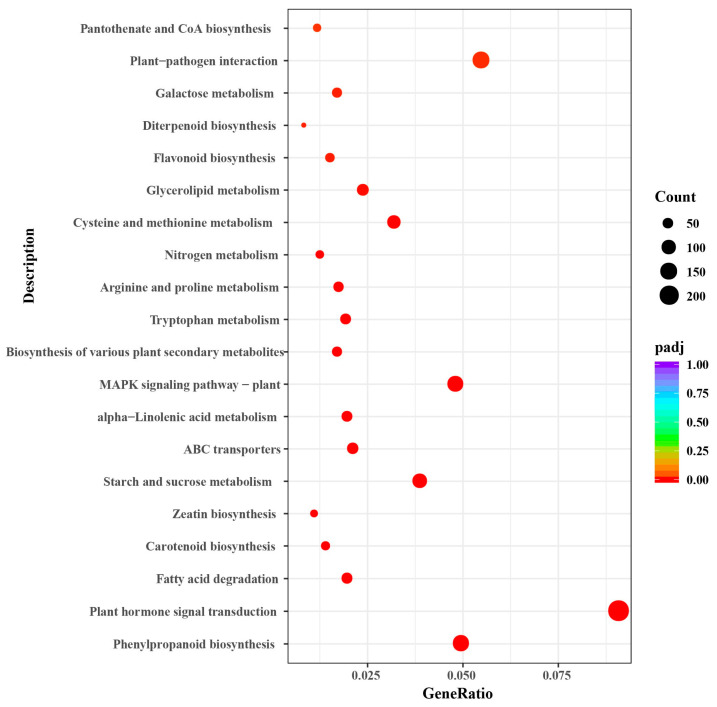
Scatter diagram of Kyoto Encyclopedia of Genes and Genomes (KEGG) pathway enrichments for genes differentially expressed in *G. anomalum* under salt stress. The Enrichment Factor represents the ratio of the proportion of differentially expressed proteins in the proportion of total *G. anomalum* proteins annotated as involved in a given pathway.

**Figure 8 plants-13-00312-f008:**
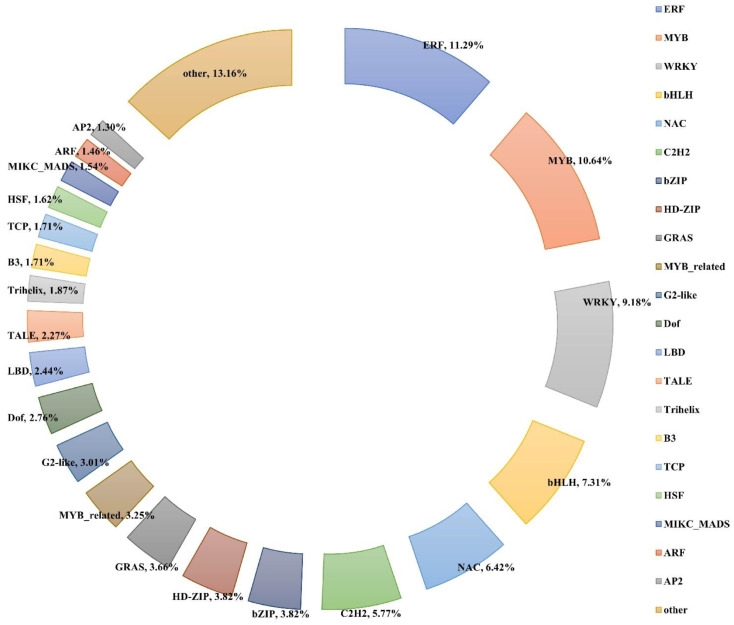
Distribution of differentially expressed TFs. Each sector represents the relative proportion of the family.

**Figure 9 plants-13-00312-f009:**
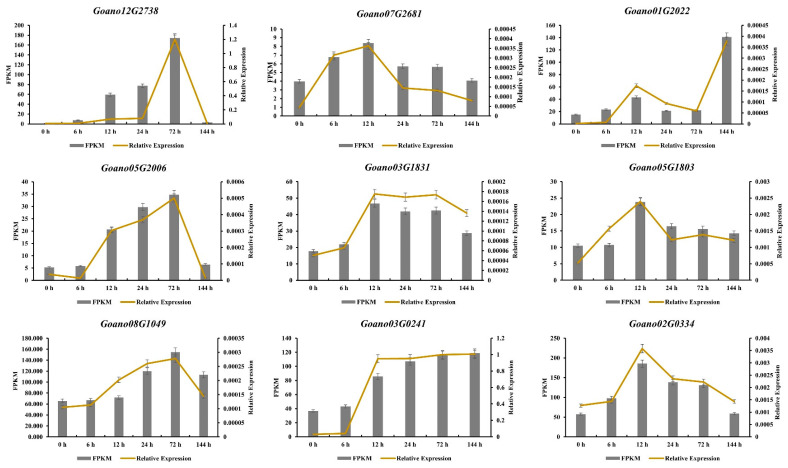
Validation of RNA-Seq expression patterns for nine genes by qRT-PCR. Bars represent the relative intensity of qRT-PCR products and the FPKM value from three independent biological replicates. The left *y*-axis shows the FPKM value obtained from RNA-Seq, and the relative expression as determined by qRT-PCR.

## Data Availability

The RNA-seq data are deposited in the BioProject database of NCBI under the accession number PRJNA697836.

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
