# Peer review of "Transcriptome Expression Profiling Reveals the Molecular Response to Salt Stress in Gossypium anomalum Seedlings"

_plants, 2024, doi:10.3390/plants13020312_

Round 1

Reviewer 1 Report

Comments and Suggestions for Authors

This manuscript described transcriptome expression profiling of the molecular response to salt Stress in Gossypium anomalum seedlings. The manuscript can be accepted to publish after minor revisions.

My detailed suggestions are as follows:

1. In “5.1. Preparation of Plant Material”, how much 350 mmol/L NaCl solution was used to water cotton plants? It should be described clearly so that the experiment can be repeated by others. 

2. In “5.7. Validation of DEGs by qRT-PCR”, the authors should describe the criteria of gene selection for qRT-PCR. Were they randomly selected, or were they selected based on their expression level in the transcriptome data, or were they selected based on their gene annotation, or other reasons?

3. In Figure 1, the Pn decreased significantly with the time increased, whereas Gs/Tr/Ci decreased at 24h but then increased as time goes on; the difference between Pn and Gs/Tr/Ci is an interesting phenomenon and needs more explanation in the Discussion. Moreover, it seems that all the other time point was compared with 0h. Can they be compared with the near time point to see the significance level?

4. The manuscript is written very well, however, there are still some grammatical mistakes or typos in the manuscript, e.g., Line 70, “These data will led to” should be “These data will lead to”. Consequently, a thorough grammar check needs to be performed. 

5. Generally the references are organized well, whereas some minor changes are needed, e.g., the journal name “Resources, Environment and Sustainability” in Line 447 is a full name, whereas other journal names are abbreviated, so a through format check should be performed on the references.

Comments on the Quality of English Language

The manuscript is written very well, however, there are still some grammatical mistakes or typos in the manuscript, e.g., Line 70, “These data will led to” should be “These data will lead to”. Consequently, a thorough grammar check needs to be performed. 

Author Response

Point-by-point responses to Reviewer #1’s comments

  1. In the Figures, A, B, C etc. should be added to facilitate reviewing this manuscript.

Response: We are grateful for your valuable suggestions, and we have added “A, B, C” in Figure 1, Figure 2, Figure 3, and Figure 5.

  1. The significant analysis were made between 0 h as control and other time points as treatments, if so, the corresponding line between control and treatment should be added individually.

Response: Thanks for your valuable suggestions, and we have added the corresponding line between control and treatment in Figure 1, Figure 2, and Figure S1.

  1. Why 350 mmol/L NaCl treatment was selected? Give your explanation.

Response: Thanks for your comments. According to our previous research (Xu et al., 2021), “the second ten seedlings with two replications were selected for 350 mmol/L NaCl treatment.” We have added the reference in the section of Materials and Methods (Line 431). Before determining the concentration, we also tested various concentrations of NaCl solution. It is believed that G. anomalum, as a salt-tolerant variety, can exhibit significant salt stress effects at 350 mmol/L NaCl, and can trigger a series of related physiological and molecular responses. By comparing the expression profile changes at this salt concentration, a better understanding of the response mechanism of G. anomalum to salt stress can be obtained.”

Reference:

Peng Xu, Qi Guo, Shan Meng, Xianggui Zhang, Zhenzhen Xu, Wangzhen Guo, and Xinlian Shen. Genome-wide association analysis reveals genetic variations and candidate genes associated with salt tolerance related traits in Gossypium hirsutum. BMC Genomics, 2021, 22:26.

  1. In the discussion section, compared the homologs in upland cotton, these candidate genes from Gossypium anomalum are potential elite genes for improving salt tolerance, give the corresponding discussion about how to use the candidate genes to improve the upland cotton salt tolerance.

Response: Thanks for your comments, and we have added the corresponding discussion about how to use the candidate genes to improve the upland cotton salt tolerance in our revised manuscript (Line 366-369). “To date, no gene related to salt tolerance has been cloned in G. anomalum. Cloning these potential elite salt tolerance genes and overexpressing them in G. hirsutum will assist in the development of new upland cotton cultivars with improved salt tolerance.”

Reviewer 2 Report

Comments and Suggestions for Authors

1In the Figures , A,B,C etc. should be added to facilitate reviewing this manuscript.
2
The significant analysis were made between 0 h as control and other time points as treatments, if so , the corresponding line between control and treatment should be added individually.

3Why 350 mmol/L NaCl treatment was selected ? Give your explanation.

4In the discussion section , compared the homologs in upland cotton, these candidate genes from Gossypium anomalum are potential elite genes for improving salt tolerance, give the corresponding discussion about how to use the candidate genes to improve the upland cotton salt tolerance.

Comments on the Quality of English Language

minor editing of english language required. 

Author Response

Point-by-point responses to Reviewer #2’s comments

This manuscript described transcriptome expression profiling of the molecular response to salt Stress in Gossypium anomalum seedlings. The manuscript can be accepted to publish after minor revisions.

My detailed suggestions are as follows:

  1. In “5.1. Preparation of Plant Material”, how much 350 mmol/L NaCl solution was used to water cotton plants? It should be described clearly so that the experiment can be repeated by others.

Response: Thanks for your comments, and we have added the volume of 350 mmol/L NaCl solution in the revised manuscript (Line 431). “Every plant of paper cup (diameter 7 cm and height 8 cm) was irrigated with 75 ml of 350 mmol/L NaCl.”

  1. In “5.7. Validation of DEGs by qRT-PCR”, the authors should describe the criteria of gene selection for qRT-PCR. Were they randomly selected, or were they selected based on their expression level in the transcriptome data, or were they selected based on their gene annotation, or other reasons?

Response: Based on the expression level in the transcriptome data and functional annotations from their orthologues in Arabidopsis, nine candidate genes potentially implicated in salt tolerance were selected to be conducted quantitative real-time PCR (qRT-PCR). These nines selected genes comprised cytochrome P450 (Goano03G0241 and Goano12G2738), a WRKY transcription factor (Goano03G1831), serine acetyltransferase (Goano01G2022), a transferase family protein (Goano05G2006), a protein with peroxidase structure domain (Goano08G1049), cellulose synthase (Goano07G2681), a 1,3-beta-glucan synthase component (Goano05G1803), and a homeobox-leucine zipper protein (Goano02G0334).

We have described the criteria of gene selection for qRT-PCR in our revised manuscript (Line 292-294).

  1. In Figure 1, the Pn decreased significantly with the time increased, whereas Gs/Tr/Ci decreased at 24h but then increased as time goes on; the difference between Pn and Gs/Tr/Ci is an interesting phenomenon and needs more explanation in the Discussion. Moreover, it seems that all the other time point was compared with 0h. Can they be compared with the near time point to see the significance level?

Response: Thanks for your comments, and we have compared the Pn, Gs, Tr, and Ci value near time point in Figure S1.

  1. The manuscript is written very well, however, there are still some grammatical mistakes or typos in the manuscript, e.g., Line 70, “These data will led to” should be “These data will lead to”. Consequently, a thorough grammar check needs to be performed.

Response: Thanks for your comments, and our manuscript have been revised by a language editor, GlassPhoenix. Also, Line 70, “These data will led to significant advances in our understanding of molecular response to salt stress in Gossypium spp. that might be valuable for breeding application.” has been changed with “These data promise to significantly advance our understanding of the molecular response to salt stress in Gossypium spp., with potential value for breeding applications.” Line 91-94.

  1. Generally the references are organized well, whereas some minor changes are needed, e.g., the journal name “Resources, Environment and Sustainability” in Line 447 is a full name, whereas other journal names are abbreviated, so a through format check should be performed on the references.

Response: Thanks for your comments, and we have changed the journal name “Resources, Environment and Sustainability” with “Resour. Environ. Sustain” in our revised manuscript (Line 558).

Reviewer 3 Report

Comments and Suggestions for Authors

A brief summary

In this manuscript, G.anomalum, a specific cotton variety with high salt stress tolerance, was used as the material. Through an in-depth analysis of transcriptional changes following exposure to salt stress, a multitude of genes and transcription factors potentially associated with salt stress were successfully identified. The functions and pathways associated with these genetic elements were thoroughly examined, and the resulting analysis data were deemed accurate and reliable. This research not only enriches our understanding of the genetic landscape underpinning salt stress response in cotton but also furnishes valuable genetic resources and data for future investigations into cotton's resilience to salt stress. The insights presented in this paper hold significant reference value for further studies in the field. This research is excellent and innovative, I support publishing it in plants.

A few minor revisions are recommended:

1. In line 36, the first appearance of "FAO" needs to be marked with the full name.

2. In line 37, the data "FAO report that soil salinity occupies more than 20% of the global irrigated area" comes from 2016 and needs to be updated.

3. In line 40-41,"and after 5-10 years of land improvement, it is then used for growing wheat, corn, and other crops." This sentence can be slightly modified to "After 5-10 years of land improvement, it can be used for growing wheat, corn, and other crops."

4. In line 86-89,"Compared to 0 h, the net photosynthetic rate (Pn, μmol m-2 s-1), stomatal conductance (Gs, μmol m-2 s-1), transpiration rate (Tr, mol m-2 s-1), and intercellular CO2 concentration (Ci, µmol mol-1) showed significantly declined (P < 0.05, P < 0.01) under 350 mmol/L NaCl treatment (Figure 1)." This sentence can be slightly modified to "Compared to the control at 0 h, the net photosynthetic rate (Pn, μmol m-2 s-1), stomatal conductance (Gs, μmol m-2 s-1), transpiration rate (Tr, mol m-2 s-1), and intercellular CO2 concentration (Ci, µmol mol-1) significantly decreased (P < 0.05, P < 0.01) in G.anomalum under 350 mmol/L NaCl treatment (Figure 1)."

5. Figure 1 and Figure 2, the title font size of the ordinate axis can be appropriately increased.

6. Why choose 350mmol/L Nacl to treat G.anomalum plants?

7. In Figure 6, some words enter the inside of the picture and need to be adjusted. "Count" should be changed to "counts".

8. The language needs to be slightly modified.

9. In line 111-112, "Each sample was represented at least 39,807,348 raw reads" can be slightly modified to "Each sample generated at least 39,807,348 raw reads".

10. In line 120-123, "Compared with the control (0 h), the differentially expressed genes (DEGs) in G.anomalum roots at 6, 12, 24, 72, and 144 h after 350 mmol/L NaCl treatment were screened according to the expression abundance of transcripts (|log2FoldChange | ≥ 1 and Padj ≤ 0.05), and a total of 15,476 DEGs were identified (Table S2). " This sentence can be modified to ""Compared to the control (0 h), differentially expressed genes (DEGs) in Gossypium anomalum roots were identified at 6, 12, 24, 72, and 144 hours after treatment with 350 mmol/L NaCl. The screening criteria were based on transcript expression abundance (|log2FoldChange| ≥ 1 and Padj ≤ 0.05), resulting in the identification of a total of 15,476 DEGs (Table S2)."

11. In line 138-140, "To characterize the biological processes affected by salt stress in G. anomalum, Gene Ontology (GO) annotation of 15,476 DEGs were performed using biological process (BP), cellular component (CC), and molecular function (MF) terms. " This sentence can be modified to "To elucidate the biological processes influenced by salt stress in Gossypium anomalum, Gene Ontology (GO) annotation was conducted for the 15,476 DEGs, incorporating terms related to biological process (BP), cellular component (CC), and molecular function (MF) ".

12. In line 178-181, "Of the 15,476 DEGs, 2,698 were annotated to 125 metabolic pathways in the Kyoto Encyclopedia of Genes and Genomes (KEGG) database, including 3,193 DEGs in metabolism, 458 in genetic information processing, 465 in environmental information processing, 153 in cellular processes, and 179 in organismal systems pathways", there are data errors, which needs to be checked and revised.

Comments on the Quality of English Language

The overall English level is very high, and only some sentences need to be revised.

Author Response

Point-by-point responses to Reviewer #3’s comments

A brief summary

In this manuscript, G. anomalum, a specific cotton variety with high salt stress tolerance, was used as the material. Through an in-depth analysis of transcriptional changes following exposure to salt stress, a multitude of genes and transcription factors potentially associated with salt stress were successfully identified. The functions and pathways associated with these genetic elements were thoroughly examined, and the resulting analysis data were deemed accurate and reliable. This research not only enriches our understanding of the genetic landscape underpinning salt stress response in cotton but also furnishes valuable genetic resources and data for future investigations into cotton's resilience to salt stress. The insights presented in this paper hold significant reference value for further studies in the field. This research is excellent and innovative, I support publishing it in plants.

A few minor revisions are recommended:

  1. In line 36, the first appearance of "FAO" needs to be marked with the full name.

Response: Thank you for your reminding. We have added the full name of “Food and Agriculture Organization of the United Nations” in our revised manuscript (Line 39).

  1. In line 37, the data "FAO report that soil salinity occupies more than 20% of the global irrigated area" comes from 2016 and needs to be updated.

Response: Thank you for your suggestion, and we have updated the sentence “FAO report that soil salinity occupies more than 20% of the global irrigated area” with “The Food and Agriculture Organization of the United Nations (FAO) reports that there are more than 833 million hectares of salt-affected soils around the globe (8.7% of the planet).” in the revised manuscript (Line 39-41).

  1. In line 40-41,"and after 5-10 years of land improvement, it is then used for growing wheat, corn, and other crops." This sentence can be slightly modified to "After 5-10 years of land improvement, it can be used for growing wheat, corn, and other crops."

Response: Thanks for your comments, and we have changed the sentence “and after 5-10 years of land improvement, it is then used for growing wheat, corn, and other crops." with “After 5-10 years, the improved land can then be used for growing wheat, corn, and other crops.” in the revised manuscript (Line 46-47).

  1. In line 86-89,"Compared to 0 h, the net photosynthetic rate (Pn, μmol m-2 s-1), stomatal conductance (Gs, μmol m-2 s-1), transpiration rate (Tr, mol m-2 s-1), and intercellular CO2 concentration (Ci, µmol mol-1) showed significantly declined (P < 0.05, P < 0.01) under 350 mmol/L NaCl treatment (Figure 1)." This sentence can be slightly modified to "Compared to the control at 0 h, the net photosynthetic rate (Pn, μmol m-2 s-1), stomatal conductance (Gs, μmol m-2 s-1), transpiration rate (Tr, mol m-2 s-1), and intercellular CO2 concentration (Ci, µmol mol-1) significantly decreased (P < 0.05, P < 0.01) in G.anomalum under 350 mmol/L NaCl treatment (Figure 1)."

Response: Thanks for your comments, and this sentence have been modified in our revised manuscript (Line 109-112). “The net photosynthetic rate (Pn, μmol m-2 s-1), stomatal conductance (Gs, μmol m-2 s-1), transpiration rate (Tr, mol m-2 s-1), and intercellular CO2 concentration (Ci, µmol mol-1) were significantly decreased (P < 0.05, P < 0.01) in salt-treated G. anomalum plants compared to the 0 h control.”

  1. Figure 1 and Figure 2, the title font size of the ordinate axis can be appropriately increased.

Response: Thanks for your comments, and we have increased the title font size of the ordinate axis in Figure 1 and Figure 2.

  1. Why choose 350mmol/L Nacl to treat G.anomalum plants?

Response: Thanks for your comments. According to our previous research (Xu et al., 2021), “the second ten seedlings with two replications were selected for 350 mmol/L NaCl treatment.” We have added the reference in the section of Materials and Methods (Line 431). Also, before determining the concentration, we tested various concentrations of NaCl solution. It is believed that G. anomalum, as a salt-tolerant variety, can exhibit significant salt stress effects at 350mmol/L NaCl, and can trigger a series of related physiological and molecular responses. By comparing the expression profile changes at this salt concentration, a better understanding of the response mechanism of G. anomalum to salt stress can be obtained.

Reference

Peng Xu, Qi Guo, Shan Meng, Xianggui Zhang, Zhenzhen Xu, Wangzhen Guo, and Xinlian Shen. Genome-wide association analysis reveals genetic variations and candidate genes associated with salt tolerance related traits in Gossypium hirsutum. BMC Genomics, 2021, 22:26

  1. In Figure 6, some words enter the inside of the picture and need to be adjusted. "Count" should be changed to "counts".

Response: Thanks for your comments, and we have changed the word “Count” with “counts” in Figure 6.

  1. The language needs to be slightly modified.

Response: our manuscript has been revised by a language editor, GlassPhoenix.

  1. In line 111-112, "Each sample was represented at least 39,807,348 raw reads" can be slightly modified to "Each sample generated at least 39,807,348 raw reads".

Response: Thanks for your comments, and we have changed the sentence “Each sample was represented at least 39,807,348 raw reads” with “Each sample generated at least 39,807,348 raw reads” in the revised manuscript (Line 143-144).

  1. In line 120-123, "Compared with the control (0 h), the differentially expressed genes (DEGs) in G.anomalum roots at 6, 12, 24, 72, and 144 h after 350 mmol/L NaCl treatment were screened according to the expression abundance of transcripts (|log2FoldChange | ≥ 1 and Padj ≤ 0.05), and a total of 15,476 DEGs were identified (Table S2). " This sentence can be modified to ""Compared to the control (0 h), differentially expressed genes (DEGs) in Gossypium anomalum roots were identified at 6, 12, 24, 72, and 144 hours after treatment with 350 mmol/L NaCl. The screening criteria were based on transcript expression abundance (|log2FoldChange| ≥ 1 and Padj ≤ 0.05), resulting in the identification of a total of 15,476 DEGs (Table S2)."

Response: Thanks for your comments, and we have changed the sentence “Compared with the control (0 h), the differentially expressed genes (DEGs) in G.anomalum roots at 6, 12, 24, 72, and 144 h after 350 mmol/L NaCl treatment were screened according to the expression abundance of transcripts (|log2FoldChange | ≥ 1 and Padj ≤ 0.05), and a total of 15,476 DEGs were identified.” with “Differentially expressed genes (DEGs) in G. anomalum roots relative to 0 h control were identified at 6, 12, 24, 72, and 144 hours of treatment with 350 mmol/L NaCl. The screening criteria were based on transcript expression abundance (|log2FoldChange| ≥ 1 and Padj ≤ 0.05), and yielded a total of 15,476 DEGs.” in the revised manuscript (Line 153-160).

  1. In line 138-140, "To characterize the biological processes affected by salt stress in G. anomalum, Gene Ontology (GO) annotation of 15,476 DEGs were performed using biological process (BP), cellular component (CC), and molecular function (MF) terms. " This sentence can be modified to "To elucidate the biological processes influenced by salt stress in Gossypium anomalum, Gene Ontology (GO) annotation was conducted for the 15,476 DEGs, incorporating terms related to biological process (BP), cellular component (CC), and molecular function (MF) ".

Response: Thanks for your comments, and we have changed the sentence “To characterize the biological processes affected by salt stress in G. anomalum, Gene Ontology (GO) annotation of 15,476 DEGs were performed using biological process (BP), cellular component (CC), and molecular function (MF) terms. ” with “To elucidate the biological processes influenced by salt stress in G. anomalum, Gene Ontology (GO) annotation was conducted for the 15,476 DEGs, including terms from the biological process (BP), cellular component (CC), and molecular function (MF).” in the revised manuscript (Line 178-180).

  1. In line 178-181, "Of the 15,476 DEGs, 2,698 were annotated to 125 metabolic pathways in the Kyoto Encyclopedia of Genes and Genomes (KEGG) database, including 3,193 DEGs in metabolism, 458 in genetic information processing, 465 in environmental information processing, 153 in cellular processes, and 179 in organismal systems pathways", there are data errors, which needs to be checked and revised.

Response: Thanks for your comments, and we have changed the sentence "Of the 15,476 DEGs, 2,698 were annotated to 125 metabolic pathways in the Kyoto Encyclopedia of Genes and Genomes (KEGG) database, including 3,193 DEGs in metabolism, 458 in genetic information processing, 465 in environmental information processing, 153 in cellular processes, and 179 in organismal systems pathways" with “Of the 15,476 DEGs, 2,698 were annotated to 125 metabolic pathways in the Kyo-to Encyclopedia of Genes and Genomes (KEGG) database, including 3193 DEGs in metabolism, 458 in genetic information processing, 465 in environmental information processing, 153 in cellular processes, and 179 in organismal systems pathways.” in the revised manuscript (Line 230-233). Because a gene may be involved in multiple pathways, the sum of DEGs counts for all pathways is greater than the number of genes annotated.

Reviewer 4 Report

Comments and Suggestions for Authors

Dear Authors,

Thank you for your excellent work. Salt stress is a traditional subject in plant growth and production; however, there are still areas that remain undiscovered. In this study, the authors investigated salt resistance candidate genes through RNA-SEQ, subsequently confirming their findings with qRT-PCR for selected genes.

While the study includes well-known parameters for salt stress evaluation, such as H2O2, MDA, PRO contents, CAT, and SOD activity, this information is not present in the introduction. I suggest that the authors add relevant information to the introduction regarding these parameters.

Regarding the salt stress treatment, how was the decision made to use 350 mmol/L NaCl as the treatment condition? Was it based on references from other studies, or did the authors experiment with various concentrations before determining that 350 mmol/L NaCl was the optimal condition?

In lines #41-45, where it is mentioned that "cotton production areas are continually moving to saline land 42 where is unsuitable for some food crops...," could you please provide citations for this information? Thank you.

Additionally, for Figures, I recommend refining it by using subfigures (e.g., Figure 1a, Figure 1b-e) for better clarity and presentation.

Best regards

Comments on the Quality of English Language

The language requires some enhancements.

Author Response

Point-by-point responses to Reviewer #4’s comments

Thank you for your excellent work. Salt stress is a traditional subject in plant growth and production; however, there are still areas that remain undiscovered. In this study, the authors investigated salt resistance candidate genes through RNA-SEQ, subsequently confirming their findings with qRT-PCR for selected genes.

  1. While the study includes well-known parameters for salt stress evaluation, such as H2O2, MDA, PRO contents, CAT, and SOD activity, this information is not present in the introduction. I suggest that the authors add relevant information to the introduction regarding these parameters.

Response: Thanks for your comments, and we have added the description of the well-known parameters for salt stress evaluation in the revised manuscript Line 76-84. “In general, when plants experience salt stress during their growth, they produce a significant amount of reactive oxygen species (ROS), which can negatively impact cellular function by damaging nucleic acids and oxidizing proteins. Furthermore, peroxidation of membrane lipids, particularly in the plasma membrane, damages the structure of the cell membrane and alters its permeability. As a byproduct of this process, malondialdehyde (MDA) is produced. To counteract the stress-induced accumulation of ROS, plants employ signal transduction and an antioxidant enzyme system that includes enzymes such as superoxide dismutase (SOD), catalase (CAT), and peroxidase (POD).”

  1. Regarding the salt stress treatment, how was the decision made to use 350 mmol/L NaCl as the treatment condition? Was it based on references from other studies, or did the authors experiment with various concentrations before determining that 350 mmol/L NaCl was the optimal condition?

Response: Thanks for your comments. According to our previous research (Xu et al., 2021), “the second ten seedlings with two replications were selected for 350 mmol/L NaCl treatment.” We have added the reference in the section of Materials and Methods (Line 431). Also, before determining the concentration, we tested various concentrations of NaCl solution. It is believed that G. anomalum, as a salt-tolerant variety, can exhibit significant salt stress effects at 350 mmol/L NaCl, and can trigger a series of related physiological and molecular responses. By comparing the expression profile changes at this salt concentration, a better understanding of the response mechanism of G. anomalum to salt stress can be obtained.”

Reference:

Peng Xu, Qi Guo, Shan Meng, Xianggui Zhang, Zhenzhen Xu, Wangzhen Guo, and Xinlian Shen. Genome-wide association analysis reveals genetic variations and candidate genes associated with salt tolerance related traits in Gossypium hirsutum. BMC Genomics, 2021, 22:26.

  1. In lines #41-45, where it is mentioned that "cotton production areas are continually moving to saline land 42 where is unsuitable for some food crops...," could you please provide citations for this information? Thank you.

Response: Thanks for your comments, and we have added the cited reference in our revised manuscript (Line 50).

Reference:

Li, W.; Wang, Z.; Zhang, J. et al. Soil salinity variations and cotton growth under long-term mulched drip irrigation in saline-alkali land of arid oasis. Irrig Sci .2022, 40, 103–113.

  1. Additionally, for Figures, I recommend refining it by using subfigures (e.g., Figure 1a, Figure 1b-e) for better clarity and presentation.

Response: Thanks for your comments, and we have added “A, B, C” in Figure 1, Figure 2, Figure 3, and Figure 5.

Round 2

Reviewer 4 Report

Comments and Suggestions for Authors

Dear Author,

Thank you for the updated version. It shows improvement, and I recommend accepting the updated version. Thank you for your efforts. 

Have a blessed day.